# Association between acute respiratory acidosis and hyperkalemia during esophageal cancer surgery in the prone position: A multicenter retrospective observational study protocol

Sakura Okamoto[ID][1]*, Tokiko Tochii[2], Jyunya Nakada[2], Hideaki Note[1]

1 Department of Anesthesiology, Aichi Medical University, Nagakute, Aichi, Japan, 2 Department of Anesthesiology, Aichi Cancer Center, Nagoya, Japan

* sokamotoanes@gmail.com

## Abstract

### Introduction

Esophagectomy in the prone position can induce acute respiratory acidosis. While it is known that metabolic acidosis can significantly elevate serum potassium, the effect of respiratory acidosis is conventionally considered to cause minimal or no elevation. However, clinical practice in this surgical setting sometimes reveals a greater-than-expected degree of hyperkalemia. The objective of this study is to investigate the association between this acute respiratory acidosis and hyperkalemia, and to explore its clinical implications.

### Methods and analysis

This is a retrospective, two-center observational study of adult patients who underwent thoracoscopic esophagectomy in the prone position at two major Japanese institutions between January 2022 and December 2024. The primary outcome is the mean difference in serum potassium levels between the supine (baseline) and prone-position phases, analyzed as a paired-design within each patient. Key secondary outcomes include: 1) a multivariable analysis to identify factors associated with the magnitude of potassium increase; 2) a sensitivity analysis of the primary outcome after excluding cases with significant metabolic acidosis (e.g., base excess $< -5$ mmol/L); and 3) correlation analyses between the change in potassium and various physiological parameters, including $PaCO_2$ and markers of metabolic stress. Analyses will include the paired t-test, correlation analyses, and multivariable regression models.

**Data availability statement:** Data cannot be shared publicly due to ethical and privacy restrictions on patient data. This protocol is for a study where data extraction has not yet begun. Upon completion of the study, anonymized data may be available from the Aichi Medical University Institutional Review Board (contact via amu_ethics@aichi-med-u.ac.jp) for qualified researchers who meet the criteria for access to confidential data. The preliminary data used for the sample size calculation is subject to these same restrictions.

**Funding:** The author(s) received no specific funding for this work.

**Competing interests:** The authors have declared that no competing interests exist.

## Introduction

Prone positioning has become an increasingly common approach for thoracoscopic esophageal cancer surgery to facilitate surgical exposure. It has been associated with better surgical outcomes and improved perioperative safety compared to the classical lateral decubitus position [1]; however, it may significantly impact ventilation. Partial lung collapse and intrathoracic insufflation of carbon dioxide ($CO_2$), commonly used in these procedures, may contribute to significant alterations in respiratory physiology and acid-base balance.

Clinically, elevations in serum potassium levels associated with acute respiratory acidosis are often observed during these surgeries. While acidosis is generally known to affect potassium distribution, respiratory acidosis is typically considered to cause only minimal or no elevation in serum potassium compared to metabolic acidosis [2–3]. However, recent clinical reports challenge this view in the setting of critical illness; for instance, Pathak and Nanda described cases of severe, refractory hyperkalemia in COVID-19 patients where acute respiratory acidosis was identified as a major contributing factor [4]. Nevertheless, the combination of prone positioning, ventilatory compromise, and $CO_2$ insufflation may alter the expected physiological response.

The primary aim of this study is to quantify the magnitude of serum potassium elevation associated with acute respiratory acidosis during prone-position esophagectomy. Furthermore, we aim to explore the physiological drivers of this phenomenon, including the respective contributions of respiratory and metabolic factors. Ultimately, this research may contribute to enhancing the safety of ventilation strategies and optimizing electrolyte management for these complex procedures.

## Methods

### Study design

This will be a retrospective, two-center observational study.

### Study status

The study period covers patients treated between January 2022 and December 2024. While the clinical records for this period already exist, data extraction for this specific research purpose, as well as all statistical analyses, have not yet begun.

We plan to commence data collection (extraction) on or after December 1, 2025, with an estimated completion by March 2026. Authors will not have access to information that could identify individual participants during or after data collection, as all data will be fully anonymized prior to analysis. We anticipate that statistical analysis will be completed and results are expected by May 2026. This protocol is being registered and published prior to data access and analysis to ensure transparency and pre-specify the analytical plan, thereby minimizing the risk of outcome reporting bias.

### Participants

All adult patients who underwent esophageal cancer surgery, including robot-assisted procedures, in the prone position between January 2022 and December 2024 at Aichi

Medical University Hospital and Aichi Cancer Center will be included. Patients with pre-existing chronic kidney disease (not requiring dialysis) are eligible for inclusion, and the influence of baseline renal function will be evaluated as a pre-planned subgroup analysis.

## Exclusion criteria

● Patients undergoing chronic dialysis ● Patients with severe electrolyte abnormalities (defined as a preoperative serum potassium level of 5.0 mmol/L or higher) ● Patients who received intraoperative massive transfusion (defined as having received a transfusion for a blood loss of 2000 mL or more).

## Data collection

The following data will be collected from medical records: ● **Patient demographics:** Age, sex, BMI, ASA PS ● **Preoperative status:** eGFR, history of diabetes, hypertension, respiratory dysfunction or heart disease, perioperative use of diuretics, insulin, or β2-agonists. ● **Procedural details:** ○ Surgical approach (e.g., thoracoscopic, robot-assisted, or open thoracotomy) ○ Anesthesia method ○ Surgical time, Anesthesia time ○ Duration in the prone position ○ $CO_2$ insufflation (presence, duration, and pressure) ○ Bronchial blocker usage ● **Hemodynamics and fluid balance:** ○ The minimum mean arterial pressure (MAP) during the main surgical phase (prone phase) ○ Intraoperative urine output, Fluid volume, Blood transfusion status ○ Norepinephrine use and total dose ● **Arterial blood gas data:** ○ Serum potassium, pH, $PaCO_2$, $HCO_3^-$, BE, AG, and lactate will be collected at the following three timepoints: ■ **Timepoint 1 (Baseline):** After induction of anesthesia, before surgical positioning or incision. ■ **Timepoint 2 (Intervention):** During the phase of maximal respiratory challenge (e.g., the point of highest $PaCO_2$ during prone positioning with capnothorax). In cases where arterial blood gas was measured multiple times during the prone phase, the data from the timepoint with the lowest arterial pH will be used for the primary analysis. ■ **Timepoint 3 (Post-procedure):** Immediately upon arrival in the ICU or at the end of surgery.

## Outcomes

**Primary outcome.** The mean difference in serum potassium levels between the baseline (supine) and the intraoperative (prone) timepoints.

**Secondary outcomes.** The secondary outcomes are designed to explore the physiological drivers of the potassium elevation and to test the robustness of the primary finding. They are categorized as follows:

1. **Analyses of Factors Associated with Potassium Increase:**

   ○ **Correlation Analysis:** To explore potential physiological drivers, correlations between the change in serum potassium ($\Delta K^+$) and changes in other variables will be assessed. These variables will include the change in $PaCO_2$ ($\Delta PaCO_2$) and markers of metabolic stress, such as the change in lactate ($\Delta Lac$).

   ○ **Multivariable Analysis of $\Delta K^+$:** A multivariable linear regression analysis will be performed to identify factors independently associated with the magnitude of the serum potassium increase ($\Delta K^+$). Candidate covariates for the multivariable model include the change in $PaCO_2$ ($\Delta PaCO_2$), change in lactate ($\Delta Lactate$), baseline eGFR, fluid balance, norepinephrine dose, and the duration of the prone position. The final selection of independent variables will be determined based on clinical relevance, assessment of multicollinearity (e.g., variance inflation factor), and the sufficiency of outcome events to avoid overfitting.

2. **Sensitivity Analysis:**

   ○ To assess the influence of significant concurrent metabolic acidosis, a sensitivity analysis of the primary outcome will be performed after excluding cases with severe metabolic acidosis (e.g., base excess <-5 mmol/L at Timepoint 2).

- To confirm the specific impact of significant respiratory acidosis, a second sensitivity analysis will be performed restricted to patients with $PaCO_2 \geq 50$ mmHg at the intervention timepoint.

3. **Exploratory and Clinical Analyses:**

- **Risk Factors for Clinical Hyperkalemia:** As an exploratory analysis, a multivariable logistic regression will be performed to identify risk factors for the development of clinically significant hyperkalemia (defined as serum potassium > 5.0 mmol/L).

- **Subgroup Analysis:** Subgroup analyses will be conducted based on key clinical factors, including **preoperative renal function (e.g., stratified by eGFR categories)** and bronchial blocker usage.

- **Assessment of Acid-Base Components:** Changes in other acid-base parameters (e.g., $HCO_3^-$, BE, anion gap) will be descriptively analyzed.

## Sample size estimation

The sample size calculation is based on the study's primary outcome: the mean difference in serum potassium levels within the prone-position cohort. Our preliminary data from 50 cases indicate that serum potassium concentration increases by a mean of 0.73 mmol/L (standard deviation [SD] 0.53) during the prone phase. While this large effect size would allow for statistical significance with a very small sample, we aimed to power the study to detect a more conservative, yet still clinically significant, difference of 0.5 mmol/L. Although the precise threshold for adverse cardiovascular effects associated with an acute potassium increase remains undefined, the relationship between serum potassium and cardiovascular events typically follows a U-shaped curve [5]. Recent evidence further suggests that the magnitude of potassium increase is a significant predictor of arrhythmias [6]. Therefore, in the clinical setting of esophagectomy, we considered a 0.5 mmol/L increase to be a clinically relevant safety margin. Assuming a paired t-test with a two-sided alpha of 0.05 and a power of 80%, a minimum of 34 patients in the prone-position cohort is required.

We plan to include all eligible patients from both institutions within the study period (January 2022 - December 2024), targeting a total sample size of approximately 150 cases. While the primary power calculation is met with 34 patients, this larger sample size will provide robust data for the secondary outcomes, including the multivariable regression analyses, ensuring greater precision and the ability to explore multiple predictor variables.

## Statistical analysis

The selection of the blood gas sample with the lowest pH during the intervention phase is a pre-specified criterion. This approach is not considered a selection bias but is rather a deliberate strategy to focus the analysis on the study's primary research question: the physiological response, particularly in potassium homeostasis, to the most severe degree of acute acidosis encountered during the procedure.

All statistical analyses will be performed using R version 4.3.2 (R Foundation for Statistical Computing, Vienna, Austria). A p-value of less than 0.05 will be considered statistically significant.

**Descriptive statistics.** Continuous variables will be presented as mean ± SD or median (interquartile range) as appropriate, while categorical variables will be presented as numbers (percentages).

**Analysis for the primary outcome.** A paired t-test will be used to compare the mean serum potassium levels between the baseline and intraoperative timepoints.

**Analyses for secondary outcomes.**

1. **Analyses of Factors Associated with Potassium Increase:** For correlation analysis, Pearson or Spearman correlation will be used. For multivariable analysis of $\Delta K^+$, a linear regression model will be developed.

2. **Sensitivity Analysis:** The primary outcome will be re-analyzed using a paired t-test on the subset of patients excluding those with severe metabolic acidosis.

3. **Exploratory and Clinical Analyses:** For identifying risk factors for clinical hyperkalemia, a multivariable logistic regression model will be used. For subgroup analyses, the primary outcome will be compared between strata (e.g., using an independent t-test or ANOVA). Other exploratory analyses will be descriptive or use appropriate statistical tests as needed.

### Ethics and dissemination

This study protocol has been approved by the institutional review board of Aichi Medical University (Approval number: 2025−088) and the Aichi Cancer Center. As this is a retrospective study using anonymized data, the requirement for individual written informed consent was waived. An opt-out policy is in place for patient notification via institutional websites. All collected data will be anonymized. The findings will be disseminated through presentations at academic conferences and publication in a peer-reviewed journal. The study protocol has been registered with the University Hospital Medical Information Network Clinical Trials Registry (UMIN-CTR: UMIN000058829).

### Discussion

This study protocol was designed to address a critical clinical observation: a greater-than-expected elevation in serum potassium during prone-position esophagectomy. This finding stands in contrast to the conventional physiological understanding that respiratory acidosis—a common consequence of this procedure—causes minimal or no elevation in serum potassium, particularly when compared to metabolic acidosis [2–3]. Therefore, the primary objective of this study is first to confirm and quantify the mean increase in serum potassium between the supine (baseline) and prone-position phases. Given that acute respiratory acidosis is an almost inevitable event during this surgical phase, demonstrating a statistically significant potassium increase would, in itself, strongly suggest a clinically relevant phenomenon that challenges existing knowledge. However, we recognize that major surgery introduces multiple confounding variables that could influence potassium levels, such as metabolic stress. To address this complexity, our secondary analyses are designed to (1) assess the robustness of our primary finding via sensitivity analysis that controls for the effect of metabolic acidosis, and (2) explore the association between the potassium increase and various physiological drivers, including the severity of respiratory acidosis itself.

In particular, surgical techniques involving partial lung collapse and intrathoracic $CO_2$ insufflation may lead to rapid external $CO_2$ loading, resulting in acid-base changes distinct from those seen in endogenous respiratory acidosis. Multiple mechanisms, including $H^+/K^+$ exchange and altered ATPase activity, may contribute to this response. The body's compensatory systems may not be able to fully respond to the acute rise in $PaCO_2$, potentially leading to more pronounced potassium fluctuations than typically expected.

The challenge of managing this surgically-induced state of respiratory compromise aligns with the central theme of our previous research: using specific, non-physiological models of difficult ventilation to elucidate fundamental truths about respiratory physiology. For instance, in our previous study, we focused on patients with severe airway stenosis undergoing airway stenting—another scenario involving extreme, surgically-created ventilatory compromise. In that setting, we challenged the conventional wisdom that muscle relaxants are contraindicated. Our previous prospective study demonstrated that controlled positive-pressure ventilation with muscle relaxants significantly reduced hypoxic events compared to maintaining spontaneous respiration [7]. That study highlighted how standard management approaches can fail in such unique circumstances. The current study extends this principle from the primary challenge of 'ventilation failure' to its secondary consequence, 'electrolyte failure'. Both studies, therefore, underscore a common lesson: the body's response under these non-physiological conditions can be far more significant than conventionally anticipated.

The potential findings of this study are expected to have direct implications for the perioperative management of patients undergoing prone-position esophagectomy. If a significant potassium increase is confirmed, it would strongly suggest that rigorous monitoring is essential. This implies that arterial blood gas analysis should be considered whenever significant hypercapnia is suspected, even in the absence of desaturation (low $SpO_2$). This is particularly crucial given the potential for a large arterial to end-tidal $CO_2$ gradient ($PaCO_2$-$ETCO_2$) in this setting, which can mask the true severity of respiratory acidosis [8]. Early detection of hyperkalemia would enable timely therapeutic interventions to mitigate the risk of adverse events. Furthermore, as this surgery involves direct manipulation in close proximity to the heart, it inherently carries a high risk of cardiac arrhythmias. The addition of acute hyperkalemia—a known pro-arrhythmic factor—could substantially amplify this risk, making its detection and management a critical safety priority.

Beyond this specific surgical procedure, our findings hold potential implications for the management of critically ill patients in the ICU, particularly those with respiratory failure. Patients with conditions such as Acute Respiratory Distress Syndrome (ARDS) often present with comorbid cardiac or renal dysfunction, making them exceptionally vulnerable to the life-threatening consequences of hyperkalemia. Our findings may suggest that hyperacute respiratory acidosis, in particular, warrants closer monitoring of electrolyte balance than previously appreciated. This perspective may also prompt a re-evaluation of permissive hypercapnia strategies. While permissive hypercapnia is employed to protect the lungs, its potential impact on electrolyte homeostasis is not fully understood.

Our exploratory multivariable analysis, by seeking to identify predictors for the magnitude of potassium increase, may help in identifying surrogate markers to define the safe limits of hypercapnia in critically ill patients.

This study has several methodological strengths. First and foremost, the use of a paired-design, where each patient serves as their own control, is a powerful approach to minimize the influence of inter-patient variability. Second, its two-center nature increases the generalizability of our findings. Furthermore, the collection of detailed, serial arterial blood gas data allows for a granular analysis of the acute physiological changes. Finally, our pre-specified plan to conduct a sensitivity analysis demonstrates a rigorous effort to isolate the effects attributable to respiratory changes, thereby strengthening the specificity of our conclusions.

Several limitations of this study should be acknowledged. First, its retrospective design leads to inherent variability in the timing of blood gas sampling and a lack of standardized protocols. Second, our study includes only patients in whom blood gas analysis was performed, potentially introducing selection bias. However, we believe this may have selectively enriched our sample with patients exhibiting more severe respiratory compromise, which aligns with our primary research question. Finally, distinguishing the precise contributions of respiratory and metabolic components to a mixed acid-base disorder is inherently challenging.

In conclusion, this study protocol describes a robust investigation into the clinically significant hyperkalemia associated with hyperacute respiratory acidosis, a phenomenon particularly relevant in modern thoracic surgery. While a randomized controlled trial comparing prone and lateral surgical approaches might have once been considered the ideal design, such a trial is likely no longer ethically or practically feasible given the established clinical benefits of the prone-position approach.

Therefore, a well-designed, multicenter observational study such as the present one represents the best available methodology to elucidate this important physiological question. The insights gained from this research have the potential to refine electrolyte management, prompt a re-evaluation of concepts like permissive hypercapnia, and ultimately improve the safety of patients both in the operating room and the intensive care unit.

## Author contributions

**Conceptualization:** Sakura Okamoto, Jyunya Nakada.

**Data curation:** Sakura Okamoto, Tokiko Tochii.

**Investigation:** Tokiko Tochii.

**Methodology:** Sakura Okamoto.

**Project administration:** Sakura Okamoto.

**Resources:** Jyunya Nakada, Hideaki Note.

**Supervision:** Jyunya Nakada, Hideaki Note.

**Writing – original draft:** Sakura Okamoto.

**Writing – review & editing:** Sakura Okamoto, Tokiko Tochii, Jyunya Nakada, Hideaki Note.

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
