## [Decision Letter · Decision Letter 0]

8 Jan 2026

Dear Dr. Sakura Okamoto,

Thank you for submitting your manuscript to PLOS ONE. After careful consideration, we feel that it has merit but does not fully meet PLOS ONE’s publication criteria as it currently stands. Therefore, we invite you to submit a revised version of the manuscript that addresses the points raised during the review process.

We look forward to receiving your revised manuscript.

Kind regards,

Paolo Aurello

Academic Editor

PLOS One

Journal Requirements:

Reviewers' comments:

Reviewer's Responses to Questions

**Comments to the Author**

1. Does the manuscript provide a valid rationale for the proposed study, with clearly identified and justified research questions?

Reviewer #1: Yes

2. Is the protocol technically sound and planned in a manner that will lead to a meaningful outcome and allow testing the stated hypotheses?

Reviewer #1: Yes

3. Is the methodology feasible and described in sufficient detail to allow the work to be replicable?

Reviewer #1: Yes

4. Have the authors described where all data underlying the findings will be made available when the study is complete?

Reviewer #1: Yes

5. Is the manuscript presented in an intelligible fashion and written in standard English?

Reviewer #1: Yes

You may also provide optional suggestions and comments to authors that they might find helpful in planning their study.

Reviewer #1: This is a well-structured and transparently reported research protocol that explores a clinically relevant and physiologically compelling question. The authors propose a retrospective, multicenter observational study to investigate the association between acute respiratory acidosis and hyperkalemia during prone esophagectomy—a phenomenon that appears to challenge traditional physiology teaching.

1. Defining the “intervention” time point: While selecting the lowest pH blood gas during the prone position is an effective and targeted strategy, pre-defining the criteria for data extraction is helpful. For example, should the lowest pH value be used initially, or should a minimum PaCO2 threshold (e.g., > 50 mmHg) be set to ensure that this time point truly represents a significant respiratory challenge? This would enhance consistency between the two centers.

2. Covariate selection for the multivariate model: To further strengthen the pre-description, consider listing candidate variables to be included in the model (e.g., ΔPaCO2, ΔLactate, baseline eGFR, fluid balance, norepinephrine dose, duration of prone position). Briefly describing the variable selection strategy (e.g., based on clinical relevance and/or univariate association with p-values < 0.1) will help further clarify the process.

3. Discussion of Confounding Factors: Briefly mention other perioperative factors that may affect potassium (such as the use of diuretics, insulin, or β2 agonists), and state that if the distribution is uneven, it will be recorded and considered as a covariate to further strengthen the causal inference.

4. Sample Size Justification: A minor supplement is to briefly explain why a difference of 0.5 mmol/L is considered clinically significant, or to link the increase in potassium to arrhythmic potentials by citing literature.

**Do you want your identity to be public for this peer review?** For information about this choice, including consent withdrawal, please see our Privacy Policy

Reviewer #1: No

---

## [Author Response · Author response to Decision Letter 1]

9 Jan 2026

Response to Reviewers

Manuscript ID: [PONE-D-25-58930]

Title: Association between acute respiratory acidosis and hyperkalemia during esophageal cancer surgery in the prone position: A multicenter retrospective observational study protocol

We would like to thank the editor and the reviewer for their time and valuable comments. We have revised the manuscript in accordance with the suggestions, which have significantly improved the quality and clarity of our protocol. Our point-by-point responses are detailed below.

Comment 1. Defining the “intervention” time point: While selecting the lowest pH blood gas during the prone position is an effective and targeted strategy, pre-defining the criteria for data extraction is helpful. For example, should the lowest pH value be used initially, or should a minimum PaCO2 threshold (e.g., > 50 mmHg) be set to ensure that this time point truly represents a significant respiratory challenge? This would enhance consistency between the two centers.

Response: We appreciate this valuable suggestion regarding the definition of the intervention timepoint. As this is a retrospective study, our priority is to analyze the overall clinical picture without reducing the sample size unnecessarily; therefore, we have chosen not to apply a strict PaCO₂ threshold for the primary analysis. However, we agree that confirming the specific impact of significant respiratory acidosis is important.

To address this, we have added a sensitivity analysis restricted to patients with PaCO₂ ≥ 50 mmHg. This allows us to verify whether the association holds true in the population with strictly defined respiratory acidosis, as you suggested. We have updated the Methods section (Secondary Outcomes) accordingly.

Comment 2. Covariate selection for the multivariate model: To further strengthen the pre-description, consider listing candidate variables to be included in the model (e.g., ΔPaCO2, ΔLactate, baseline eGFR, fluid balance, norepinephrine dose, duration of prone position). Briefly describing the variable selection strategy (e.g., based on clinical relevance and/or univariate association with p-values < 0.1) will help further clarify the process.

Response: Thank you for this constructive comment. We agree that clarifying the variable selection process is essential.

As suggested, we have explicitly listed the candidate covariates in the Methods section (Statistical Analysis). Regarding the variable selection strategy, we have adopted the approach based on clinical relevance and the sufficiency of outcome events to prevent overfitting, as mentioned in your comment.

Comment 3. Discussion of Confounding Factors: Briefly mention other perioperative factors that may affect potassium (such as the use of diuretics, insulin, or β2 agonists), and state that if the distribution is uneven, it will be recorded and considered as a covariate to further strengthen the causal inference.

Response: We agree with your insightful suggestion. Perioperative medications such as diuretics, insulin, and β2-agonists can indeed influence potassium levels.

Accordingly, we have added these medications to our data collection and will assess their association with the outcome in the univariate analysis. However, since our study population primarily consists of patients undergoing cancer resection who generally lack significant comorbidities (other than the malignancy), we anticipate that the number of patients using these medications may be very small.

If the number of cases is insufficient to include these variables in the multivariable model, we will report the descriptive data and address this as a limitation in the Discussion section. We have updated the Methods section (Data Collection) to reflect this addition.

Comment 4. Sample Size Justification: A minor supplement is to briefly explain why a difference of 0.5 mmol/L is considered clinically significant, or to link the increase in potassium to arrhythmic potentials by citing literature.

Response: We appreciate this opportunity to clarify our rationale. While the exact threshold at which an acute potassium increase becomes pro-arrhythmic is not definitively established, meta-analyses have demonstrated a U-shaped relationship between serum potassium levels and cardiovascular risk [Ref 5]. Moreover, recent studies emphasize that the magnitude of potassium change is a critical predictor of electrocardiographic abnormalities [Ref 6].

In the specific context of esophagectomy, which involves the prone position and surgical manipulation near the heart, we determined that capturing a 0.5 mmol/L increase is clinically meaningful to identify patients at potential risk. We have added this rationale and cited the relevant literature in the Methods section (Sample Size Calculation).

---

## [Editor Report · Decision Letter 1]

14 Jan 2026

Association between acute respiratory acidosis and hyperkalemia during esophageal cancer surgery in the prone position: A multicenter retrospective observational study protocol

PONE-D-25-58930R1

Dear Dr. Sakura Okamoto,

We’re pleased to inform you that your manuscript has been judged scientifically suitable for publication and will be formally accepted for publication once it meets all outstanding technical requirements.

Kind regards,

Paolo Aurello

Academic Editor

PLOS One

---

## [Editor Report · Acceptance letter]

PONE-D-25-58930R1

PLOS One

Dear Dr. Okamoto,

I'm pleased to inform you that your manuscript has been deemed suitable for publication in PLOS One. Congratulations! Your manuscript is now being handed over to our production team.

Kind regards,

on behalf of

Dr. Paolo Aurello

Academic Editor

PLOS One